# CONNECTIONS BETWEEN SCHEDULE-FREE OPTIMIZERS, ADEMAMIX, AND ACCELERATED SGD VARIANTS

## ABSTRACT

Recent advancements in deep learning optimization have introduced new algorithms, such as Schedule-Free optimizers, AdEMAMix, MARS and Lion which modify traditional momentum mechanisms. In a separate line of work, theoretical acceleration of stochastic gradient descent (SGD) in noise-dominated regime has been achieved by decoupling the momentum coefficient from the current gradient's weight. In this paper, we establish explicit connections between these two lines of work. We substantiate our theoretical findings with experiments on 300m and 150m scale language modeling task. We find that AdEMAMix, which most closely resembles accelerated versions of stochastic gradient descent, exhibits superior performance. Building on these insights, we introduce a modification to AdEMAMix, termed Simplified-AdEMAMix, which maintains the same performance as AdEMAMix across both large and small batch-size settings while eliminating the need for two different momentum terms. The code for Simplified-AdEMAMix is available on the repository: https://anonymous.4open.science/r/Sim-AdEMAMix-072E.

## 1 INTRODUCTION

Recently, numerous optimization algorithms have been introduced for deep learning such as Lion (Chen et al., 2023), ScheduleFreeSGD/AdamW (Defazio et al., 2024), and AdEMAMix (Pagliardini et al., 2024). While these optimizers have been proposed with distinct motivations, they share a common characteristic: each modifies the momentum scheme employed in optimization.

A separate body of theoretical research has focused on accelerating gradient descent in noisy environments. Although classical momentum methods, such as heavy-ball or Nesterov momentum, are sufficient to accelerate deterministic gradient descent (particularly for quadratic functions), they do not accelerate SGD (Jain et al., 2018; Liu & Belkin, 2020). This limitation has led to the development of alternative momentum schemes aimed at achieving acceleration in the presence of noise(Jain et al., 2018; Vaswani et al., 2019; Liu & Belkin, 2020; Gupta et al., 2023). Notably, all proposed accelerated SGD methods can be interpreted as decoupling the momentum coefficient from the weight assigned to the current gradient in the optimizer update.

Our primary contribution is to establish a direct connection between the ideas developed in these two research directions. Specifically, we demonstrate that Schedule-Free SGD is mathematically equivalent to performing accelerated SGD followed by weight averaging. Furthermore, optimizers such as Lion, Schedule-Free AdamW, and AdEMAMix can be understood as combining preconditioning techniques with accelerated SGD approaches. While certain aspects of these connections have been noted in prior literature (Defazio, 2021), to the best of our knowledge, the relationship between these recently proposed optimizers and accelerated SGD has not been formally established before.

To validate our theoretical findings, we conduct experiments using a 300m and a 150m decoder-only transformer model, trained on 6b and 15b tokens respectively with a small batch size of 32k tokens, ensuring that the training process operates in a noise-dominated regime. As predicted by our theoretical insights, the performance of Schedule-Free AdamW closely aligns with that of accelerated SGD-based AdamW (Algorithm 3). Additionally, we observe that accelerated methods offer slightly improved performance at small batch sizes. However, we also demonstrate that these performance

benefits diminish at sufficiently large batch sizes, which is consistent with the theoretical connections to accelerated SGD.

Our main contributions are stated below:

1. We establish precise theoretical connections between accelerated SGD and recently proposed optimizers, such as Schedule-Free SGD and AdEMAMix.

2. We provide empirical validation through experiments on a 300m and 150m decoder-only transformer, comparing AdamW, Schedule-Free AdamW, AdEMAMix, and MARS. Our findings indicate that AdEMAMix, which most closely aligns with accelerated SGD variants, demonstrates superior performance among these methods.

3. As anticipated from its equivalence to accelerated SGD, the performance advantages of these methods diminish at large batch sizes relative to Adam. Notably, we show that Adam with momentum scheduling can match the performance of AdEMAMix.

4. At high batch sizes, we observe that Schedule-Free AdamW performs significantly worse than AdamW with cosine decay, which we attribute to the intrinsic coupling of momentum and weight averaging coefficients in Schedule-Free optimizers.

5. We introduce a modification to AdEMAMix, termed Simplified-AdEMAMix, which preserves the performance of AdEMAMix across both large and small batch size regimes, while eliminating the need for two distinct momentum terms.

## 2 RELATED WORK

We review the existing literature on accelerated SGD variants and optimization algorithms that are directly relevant to our work.

Jain et al. (2018) introduced an accelerated SGD variant that demonstrated improved convergence rates for the least-squares problem. Kidambi et al. (2018) further simplified the update rule for this variant and formally established that momentum does not provide acceleration for least squares regression. Subsequent works (Liu & Belkin, 2020; Vaswani et al., 2019; Gupta et al., 2023) extended these results to general convex and strongly convex functions under various theoretical assumptions.

Over the years, several optimizers have been proposed that exhibit similarities to the accelerated SGD variants described above. Lucas et al. (2019) proposed aggregated momentum that incorporates a weighted sum of multiple momentum terms, each with distinct coefficients, to compute the final update. Ma & Yarats (2019) proposed quasi-hyperbolic momentum explicitly inspired by the theoretical framework established in Jain et al. (2018). More recently, Chen et al. (2023) proposed an optimizer called Lion discovered via a genetic search algorithm, which, similar to previous accelerated SGD variants, assigns different weights to the gradient and the momentum coefficient in the update step. Additionally, Pagliardini et al. (2024) introduced the optimizer AdEMAMix that blends two distinct momentum scales in the final update.

## 3 BACKGROUND

### 3.1 MOMENTUM

Momentum is a well-established technique for accelerating the convergence of gradient descent in deterministic settings. The momentum update for weights $w_t$, with a momentum coefficient $\beta$, is given by:

$$m_t = \beta m_{t-1} + \nabla f(w_t); \quad w_t = w_{t-1} - \eta m_t$$

### 3.2 WEIGHT AVERAGING

Weight averaging is a widely used technique in stochastic optimization to reduce noise in the iterates. Instead of returning the final iterate $w_T$, a weighted average $\bar{w}_T$ of the iterates is computed, where the weights are denoted by $\gamma_t$:

$$\bar{w}_T = (1 - \gamma_T)\bar{w}_{T-1} + \gamma_T w_T$$

All instances of weight averaging in this paper utilize coefficients $\gamma_t$ of the form $\gamma_t \approx 1 - \frac{1}{\delta t}$ for some constant $0 \le \delta \le 1$.

### 3.3 ACCELERATED SGD

In this section, we provide a generalized framework encompassing many accelerated SGD methods:

$$m_t = \beta_{a,t} m_{t-1} + g_t, \quad w_{t+1} = w_t - \eta_{a,t} m_t - \alpha_{a,t} g_t \tag{1}$$

where $\beta_{a,t}, \alpha_{a,t}, \eta_{a,t}$ are (possibly time-dependent) scalar coefficients, and $g_t$ represents the stochastic gradient evaluated at $w_t$. We use the subscript 'a' to indicate coefficients that adhere to this specific accelerated SGD formulation.

We first note that setting $\alpha_{a,t} = 0$ recovers standard SGD with momentum. Additionally, as observed in prior work, many accelerated SGD algorithms proposed in the literature—such as those introduced by Jain et al. (2018); Vaswani et al. (2019); Liu & Belkin (2020); Gupta et al. (2023)—fall directly within this framework. A precise demonstration of this equivalence is provided in Appendix B.

## 4 CONNECTIONS BETWEEN EXISTING OPTIMIZERS AND ACCELERATED SGD

In this section, we theoretically establish precise connections between existing optimizers, such as Schedule-Free optimizers and AdEMAMix, and accelerated SGD. Based on these insights, we propose a simplified variant of AdEMAMix that utilizes a single momentum term while maintaining performance comparable to AdEMAMix across both small and large batch size regimes.

### 4.1 SCHEDULE-FREE SGD

Schedule-Free SGD (Defazio et al., 2024) is a recently introduced constant learning rate optimizer designed to eliminate the need for scheduling. Following the notation used in Defazio et al. (2024), the update equations are given by:

$$y_t = (1 - \beta)z_t + \beta x_t$$
$$z_{t+1} = z_t - \gamma g(y_t)$$
$$x_{t+1} = (1 - c_{t+1})x_t + c_{t+1} z_{t+1}$$

Here, $y_t$ represents the current model weights (where the gradient is evaluated), while $x_t$ denotes the weights used for evaluation.

We first express the update in terms of $y_t$ and $m_t$, where we define $m_{t+1} = \frac{x_t - z_{t+1}}{\gamma}$.

Further simplifying $m_{t+1}$, we obtain:

$$m_t = \frac{x_t - z_{t+1}}{\gamma} \tag{2}$$

$$= \frac{x_t + \gamma g_t - z_t}{\gamma} \tag{3}$$

$$= \frac{(1 - c_t)(x_{t-1} - z_t) + \gamma g_t}{\gamma} \tag{4}$$

$$= (1 - c_t)m_{t-1} + g_t. \tag{5}$$

Thus, $m_t$ follows the momentum update in Equation (1) with $\beta_{a,t} = 1 - c_t$. Given $m_t$, we now examine the update for $y_t$:

$$y_{t+1} = (1 - \beta)z_{t+1} + \beta x_{t+1} \tag{6}$$
$$= (1 - \beta)(z_t - \gamma g_t) + \beta((1 - c_{t+1})x_t + c_{t+1}z_{t+1}) \tag{7}$$
$$= (1 - \beta)z_t + \beta x_t - (1 - \beta)\gamma g_t + \beta c_{t+1}(z_{t+1} - x_t) \tag{8}$$
$$= y_t - \gamma[\beta c_{t+1}m_t + (1 - \beta)g_t]. \tag{9}$$

Thus, $y_t$ follows the weight update in Equation (1) with $\eta_{a,t} = \gamma\beta c_{t+1}$ and $\alpha_{a,t} = \gamma(1 - \beta)$, where $w_t = y_t$. Consequently, $y_t$ in Schedule-Free SGD precisely follows the accelerated SGD framework. However, $x_t$ is used for evaluation in Schedule-Free SGD. We now analyze the dynamics of $x_t$:

$$x_{t+1} = (1 - c_{t+1})x_t + c_{t+1}z_{t+1}$$
$$x_{t+1} = (1 - c_{t+1})x_t + c_2\left(\frac{y_{t+1} - \beta x_{t+1}}{1 - \beta}\right)$$
$$x_{t+1}(1 - \beta + c_{t+1}\beta) = (1 - c_{t+1})(1 - \beta)x_t + c_{t+1}y_{t+1}$$
$$x_{t+1} = \frac{(1 - c_{t+1})(1 - \beta)x_t + c_{t+1}y_{t+1}}{(1 - c_{t+1})(1 - \beta) + c_{t+1}}.$$

Thus, $x_t$ is a weighted average of $y_t$. Recursively expanding $x_t$ confirms that it is an exponential average of $y_t$ when $c_t$ is a constant. This establishes that Schedule-Free SGD can be understood as accelerated SGD followed by weight averaging.

The benefits of Schedule-Free SGD can be attributed to two key components:

1. Improved performance compared to standard SGD with momentum, due to its equivalence to accelerated SGD.

2. The ability to use a constant learning rate without scheduling, enabled by weight averaging (specifically, tailed weight averaging; see Section 4.1.3).

We note two advantages unique to Schedule-Free SGD/Adam:

- It does not require additional memory for weight averaging.
- It eliminates the need for an explicit weight averaging coefficient as a hyperparameter.

However, in Section 5.1, we demonstrate that this coupling of momentum and weight averaging coefficients does not scale well for large batch sizes.

### 4.1.1 CASE: $\beta = 0.0$

As noted in (Defazio et al., 2024), when $\beta = 0$, Schedule-Free SGD reduces to standard SGD with weight averaging. Since $c_t = 1/t$, it applies weight averaging from the beginning.

### 4.1.2 CASE: $\beta = 1.0$

As noted in (Defazio et al., 2024), when $\beta = 1$, Schedule-Free SGD reduces to standard momentum SGD, with the momentum coefficient $\beta_{a,t}$ scaling as $1 - 1/t$.

### 4.1.3 CASE: $\beta = 0.9$

For $\beta = 0.9$, the default setting in Schedule-Free SGD:

- As $c_t$ scales as $1/t$, momentum grows as $1 - 1/t$.
- The ratio of the weight assigned to the current gradient versus momentum is fixed at $(1 - \beta)/(\beta c_{t+1}) \approx 0.11$.
- Weight averaging is applied approximately over the most recent $10\%$ of the iterates.

## 4.2 LION

The update rule for Lion (Chen et al., 2023) is given by:

$$m'_t = \beta_1 m_{t-1} + (1 - \beta_1)g_t$$
$$\theta_t = \theta_{t-1} - \eta\mathrm{sign}(m'_t)$$
$$m_t = \beta_2 m_{t-1} + (1 - \beta_2)g_t.$$

Lion (Chen et al., 2023) can be directly interpreted as an accelerated SGD method followed by a coordinate-wise sign operation.

## 4.3 MARS

In this section, we demonstrate that the practical version of the recently proposed optimizer MARS (Yuan et al., 2024), referred to as MARS-Approx, follows the accelerated SGD framework, supplemented by a preconditioning step. The update equations (ignoring bias correction and clipping) are given by:

$$c_t = g_t + \gamma\frac{\beta_1}{1 - \beta_1}[g_t - g_{t-1}]$$
$$m_t = \beta_1 m_{t-1} + (1 - \beta_1)c_t$$
$$v_t = \beta_2 v_{t-1} + (1 - \beta_2)c_t^2$$
$$x_{t+1} = x_t - \eta\frac{m_t}{\sqrt{v_t} + \epsilon}$$

where $m_t$ and $v_t$ represent the first- and second-order momentum terms, respectively, and $x_t$ denotes the model parameters. Rewriting the update using $\hat{m}_t = m_t - \gamma g_t$, we obtain:

$$c_t = g_t + \gamma\frac{\beta_1}{1 - \beta_1}[g_t - g_{t-1}]$$
$$\hat{m}_t = \beta_1 \hat{m}_{t-1} + (1 - \beta_1)(1 - \gamma)g_t$$
$$v_t = \beta_2 v_{t-1} + (1 - \beta_2)c_t^2$$
$$x_{t+1} = x_t - \eta\frac{\hat{m}_t + \gamma g_t}{\sqrt{v_t} + \epsilon}$$

This formulation illustrates that the momentum update follows the general accelerated SGD framework. However, it is important to note that MARS employs a distinct preconditioning approach compared to AdamW. We further analyze its empirical performance in Section 5.

## 4.4 ADEMAMIX

The recently proposed optimizer AdEMAMix (Pagliardini et al., 2024) shares structural similarities with accelerated SGD-based AdamW. However, instead of using a linear combination of the current gradient and the momentum term as in accelerated SGD, AdEMAMix maintains two distinct momentum terms with different coefficients and computes their linear combination. The algorithm is formally stated in Algorithm 1.

To simplify our analysis, we consider a variant of AdEMAMix with $\beta_1 = 0$. As demonstrated in Pagliardini et al. (2024), this simplified version achieves performance nearly equivalent to the full version for small batch sizes. Our experiments in Section 5 corroborate this finding. With $\beta_1 = 0$, AdEMAMix aligns with the general accelerated SGD framework (Equation (1)). Furthermore, we show that the prescribed schedules for $\beta_3$ (momentum coefficient) and $\alpha$ (which controls the relative weight assigned to the current gradient) in AdEMAMix closely match theoretical schedules proposed for accelerated SGD (Gupta et al., 2023).

In smooth convex optimization, achieving acceleration in stochastic settings requires a momentum scheme of the form:

$$\beta_{a,t} = 1 - \frac{k}{t}$$

for some constant $k > 0$, as established by Gupta et al. (2023). The AdEMAMix optimizer approximately follows this scheme by scaling up $\beta_3$ accordingly.

Additionally, note that in accelerated SGD schemes, momentum is maintained in the standard form:

$$m_t = \beta_{a,t} m_{t-1} + g_t$$

whereas in Algorithm 1, the momentum update follows:

$$m_2^{(t)} \leftarrow \beta_3^{(t)} m_2^{(t-1)} + (1 - \beta_3^{(t)}) g^{(t)}.$$

For $\beta_{a,t}$ scaling as $1 - 1/t$, the accumulated contribution of past gradients in $m_t$ in accelerated SGD grows proportionally to $t$. Similarly, the coefficient $\alpha$ in AdEMAMix also scales proportionally to $t$. Due to these similarities, AdEMAMix demonstrates improved empirical performance relative to other optimizers, as observed in Figure 1.

For large batch sizes, however, AdEMAMix exhibits a performance decline when using $\beta_1 = 0.0$, as reported in Pagliardini et al. (2024). Gupta et al. (2023) suggests that for large batch sizes, the weight assigned to the current gradient in the update must decrease. In contrast, AdEMAMix maintains a fixed weight of 1 on the current gradient, which likely contributes to its diminished performance at large batch sizes.

In the following section, we introduce a simplified variant of AdEMAMix that incorporates a weight on the current gradient, removes the need for scheduling $\alpha$ and maintains only a single momentum term. We empirically validate that this simplified version performs comparably to AdEMAMix across both small and large batch setups.

---

**Algorithm 1** Single step of AdEMAMix optimizer.

---

1: **Input:** Data distribution $\mathcal{D}$. Initial model parameters $\theta^{(0)}$. Number of iterations $T$. Learning rate $\eta$. $\epsilon$ a small constant. AdamW parameters: $\beta_1$, $\beta_2$. AdEMAMix parameters $\beta_3$, $\alpha$. Warmup parameter $T_{\alpha,\beta_3}$, note that we usually set it to $T$. $\beta_{\text{start}}$ is usually set to $\beta_1$.

2: Optional: use schedulers $\eta^{(t)}$, $\beta_3^{(t)} \leftarrow f_{\beta_3}(t, \beta_3, \beta_{\text{start}}, T_{\alpha,\beta_3})$ and $\alpha^{(t)} \leftarrow f_\alpha(t, \alpha, T_{\alpha,\beta_3})$

3: Sample batch: $x \sim \mathcal{D}$

4: Compute gradient: $g^{(t)} \leftarrow \nabla_\theta \mathcal{L}_{\theta^{(t-1)}}(x)$

5: Update the fast EMA $m_1$: $m_1^{(t)} \leftarrow \beta_1 m_1^{(t-1)} + (1 - \beta_1) g^{(t)}$

6: Update the slow EMA $m_2$: $m_2^{(t)} \leftarrow \beta_3^{(t)} m_2^{(t-1)} + (1 - \beta_3^{(t)}) g^{(t)}$

7: Update the second moment estimate: $\nu^{(t)} \leftarrow \beta_2 \nu^{(t-1)} + (1 - \beta_2) \left(g^{(t)}\right)^2$

8: Update parameters: $\theta^{(t)} \leftarrow \theta^{(t-1)} - \eta^{(t)} \left( \frac{\hat{m}_1^{(t)} + \alpha^{(t)} m_2^{(t)}}{\sqrt{\hat{\nu}^{(t)}} + \epsilon} \right)$

---

### 4.5 SIMPLIFIED ADEMAMIX

Building on the insights discussed above, we propose a simplified optimizer that eliminates the need for maintaining two separate momentum terms and removes the requirement for scheduling $\alpha$. The optimizer is formally presented in Algorithm 2, where we employ theory-style momentum (instead of the exponential moving average (EMA) style). In the final update, we assign a fixed weight $\alpha$ to the gradient. We note that setting $\alpha = 0$ recovers the standard Adam optimizer (subject to appropriate transformations of $\eta$ and $\beta_1$). In Section 5, we demonstrate that this simplified variant matches the performance of AdEMAMix across both small and large batch sizes.

## 5 EXPERIMENTS

---

**Algorithm 2** Single step of Simplified AdEMAMix optimizer.

---

1: **Input:** Data distribution $\mathcal{D}$. Initial model parameters $\theta^{(0)}$. Number of iterations $T$. Learning rate $\eta$. $\epsilon$ a small constant. AdamW parameters: $\beta_1, \beta_2$. AdEMAMix parameters $\alpha, \beta_{\text{start}}$. Warmup parameter $T_{\beta_1}$, note that we usually set it to $T$.

2: Optional: use schedulers $\eta^{(t)}, \beta_1^{(t)} \leftarrow f_{\beta_1}(t, \beta_1, \beta_{\text{start}}, T_{\beta_1})$

3: Sample batch: $x \sim \mathcal{D}$

4: Compute gradient: $g^{(t)} \leftarrow \nabla_\theta \mathcal{L}_{\theta^{(t-1)}}(x)$

5: Update the EMA $m_1$: $m_1^{(t)} \leftarrow \beta_1 m_1^{(t-1)} + g^{(t)}$

6: Update the second moment estimate: $\nu^{(t)} \leftarrow \beta_2 \nu^{(t-1)} + (1 - \beta_2) \left(g^{(t)}\right)^2$

7: Update parameters: $\theta^{(t)} \leftarrow \theta^{(t-1)} - \eta^{(t)} \left( \frac{m_1^{(t)} + \alpha g^{(t)}}{\sqrt{\hat{\nu}^{(t)}} + \epsilon} \right)$

---

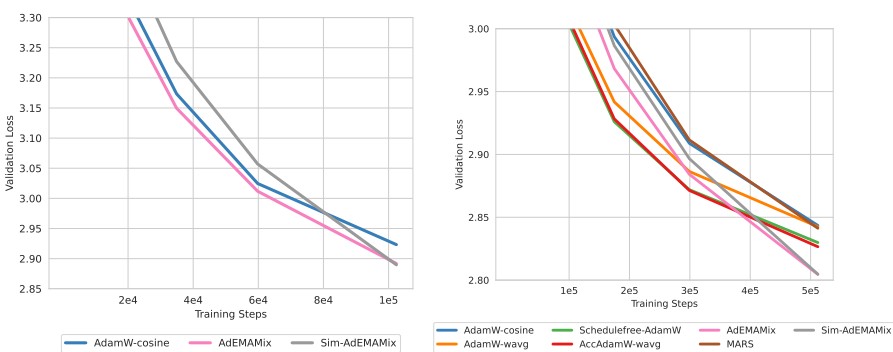

Figure 1: Comparison of the best runs of various optimizers as stated in Section 5 for language modeling task on a decoder-only 300m (**left**) and 150m (**right**) transformer model. We find that AdEMAMix and simplified-AdEMAMix perform the best, owing to their precise similarity to accelerated SGD variants.

In this section, we present experiments conducted on a 300m and 150m scale decoder-only transformer model for a language modeling task using the C4 dataset. The models are trained with a sequence length of 1024 and a batch size of 32 for around 6b tokens for 300m ($\approx 1\times$ Chinchilla) and over 15 billion tokens for 150m ($\approx 5\times$ Chinchilla), ensuring that the training operates in a noise-dominated regime.

We compare the following optimization algorithms [1]:

1. Standard AdamW with cosine decay

2. Standard AdamW with weight averaging

3. Schedule-Free AdamW

4. Accelerated AdamW with weight averaging (Algorithm 3)

5. MARS

6. AdEMAMix

7. Simplified-AdEMAMix

Details of hyperparameter sweeps for these algorithms are provided in Appendix A.

As illustrated in Figure 1, Schedule-Free AdamW and Accelerated AdamW with tailed weight averaging perform comparably, supporting our theoretical claims. Furthermore, both outperform AdamW with cosine decay and AdamW with tailed weight averaging. Moreover, AdEMAMix and Simplified-AdEMAMix outperform all methods, which we hypothesize is due to their alignment with accelerated SGD variants.

---

[1]Due to computational constraints, only a limited algorithms were compared on the 300m scale

---

**Algorithm 3** Single step of accelerated SGD based Adam with weight averaging. For simplicity we ignore the initialization, other boundary effects such as bias correction, and weight decay. Hyperparameters: Learning rate $\eta$, betas $= (\beta_1, \beta_2, \beta_3)$, weight averaging coefficient $\delta$, and epsilon $\epsilon$.

---

1: Sample batch $B_t$.
2: $g \leftarrow -\nabla_w \phi_{B_t}(w_t)$
3: $v \leftarrow \beta_2 v + (1 - \beta_2)(g \odot g)$
4: $N \leftarrow \frac{\beta_3 m + (1 - \beta_3) g}{\sqrt{\hat{v}} + \epsilon}$
5: $w \leftarrow w - \eta N$
6: $m \leftarrow \beta_1 m + (1 - \beta_1) g$
7: $c = \max(1 - 1/t, 1 - 1/(\delta t))$
8: $w_{\text{avg}} \leftarrow c w_{\text{avg}} + (1 - c) w$

---

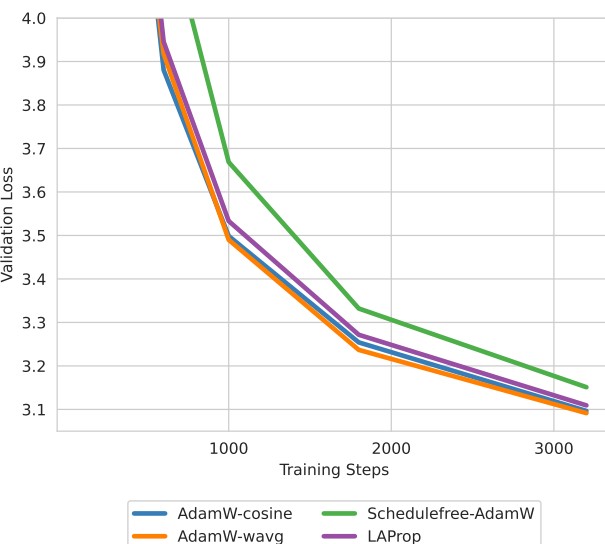

Figure 2: Comparison of the best runs of AdamW with cosine decay, schedule free AdamW, LAProp and AdamW with weight averaging at higher batch size. Experimental details can be found in Section 5.1

### 5.1 Large Batch Size Experiments

While the previous experiments focused on the small batch size regime (i.e., training with noisy gradients), we now conduct experiments in the large batch size regime to assess whether these algorithms generalize effectively. In this setup, we train the 300m and 150m model with a batch size of 1 million tokens over 6b and 3b tokens ($\approx$ Chinchilla scale) respectively.

**Schedule-Free AdamW:** As shown in Figure 2, Schedule-Free AdamW performs significantly worse compared to AdamW. We attribute this performance gap to the coupling between weight averaging and momentum coefficients. At higher batch sizes, the optimal momentum value is significantly lower than $1 - 1/t$. Although one could use a scaling factor $\approx 1 - r/t$ for some $r \geq 1$, a higher $r$ reduces the effective weight averaging window. We note that the decrease in performance of Schedule-Free optimizers at higher batch sizes was also observed in Zhang et al. (2025).

Another key distinction between AdamW and Schedule-Free AdamW is the order in which momentum and preconditioning are applied. AdamW applies momentum before preconditioning, whereas Schedule-Free AdamW applies preconditioning before momentum, making it algorithmically similar to LAProp (Ziyin et al., 2021). However, as shown in Figure 2, the performance of AdamW is comparable to that of LAProp suggesting that this difference is not the primary cause of the performance gap. Moreover, AdamW with independent coefficient for weight averaging performs

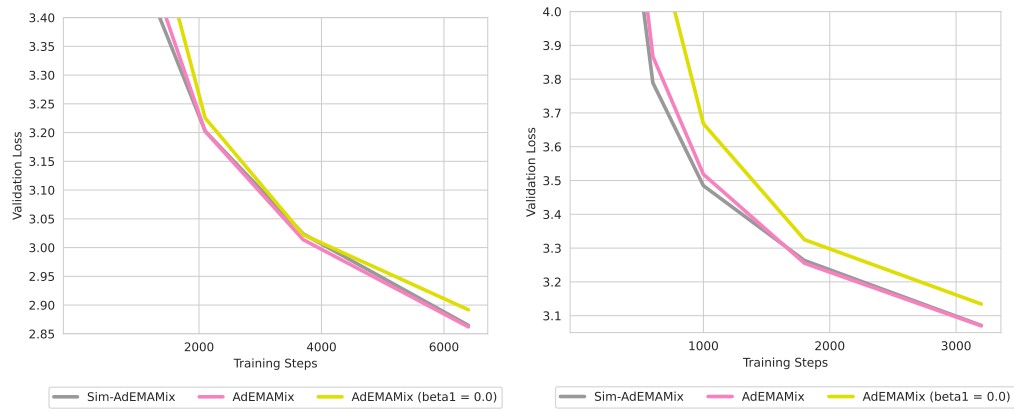

Figure 3: Comparison of the best runs of AdEMAMix (with and without $\beta_1 = 0.0$) and our variant of simplified AdEMAMix for higher batch size experiments for 300m (**Left**) and 150m(**Right**) model scale. Experimental details can be found in Section 5.1

comparably, showing that the coupling of the coefficients is the primary reason for Schedule-Free not working well at high batch sizes.

**AdEMAMix:**  For large batch sizes, as previously observed in Pagliardini et al. (2024), Figure 3 shows that setting $\beta_1 = 0.0$ in AdEMAMix results in a significant performance drop compared to using two separate momentum terms. This degradation occurs because AdEMAMix assigns a fixed weight of 1 to the current gradient, whereas theoretical accelerated SGD variants (Gupta et al., 2023) require a diminishing weight on the current gradient as batch size increases.

Additionally, as depicted in Figure 3, our proposed variant, Simplified-AdEMAMix, achieves performance equivalent to AdEMAMix while eliminating the need for two separate momentum terms. Notably, we achieve this performance at $\alpha = 0.0$, meaning Simplified-AdEMAMix reduces to standard Adam with momentum scheduling.

## 6 CONCLUSION

In this work, we establish explicit connections between accelerated SGD variants and several recently proposed optimizers, including Schedule-Free optimizers, AdEMAMix, MARS, and Lion. We also present empirical evidence demonstrating that AdEMAMix, which aligns most closely with theoretical accelerated SGD variants, achieves superior performance in small batch size training.

Building on this connection, we introduce Simplified-AdEMAMix, which removes the need for maintaining two separate momentum buffers. We empirically show that Simplified-AdEMAMix matches the performance of AdEMAMix across both small and large batch sizes while eliminating the additional memory overhead associated with AdEMAMix.

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

## A  HYPERPARAMETERS

Below are the hyperparameters for the small batch experiments.

1. AdamW with cosine decay - 51.2k warmup - learning rate in [3.16e-4, 1e-3, 3.16e-3], $\beta_1$ in [0.9, 0.95], $\beta_2$ in [0.99, 0.999, 0.99968, 0.9999]. The optimal values of $\beta_1$ and $\beta_2$ were .9 and .999 respectively matching the default values. We note that for larger batch sizes it is common to use $\beta_2 = .95$, the benfit of higher $\beta_2$ at smaller batch sizes has also been observed by Porian et al. (2024).

2. AdamW with cosine decay - 10k warmup - learning rate in [3.16e-4, 1e-3, 3.16e-3], $\beta_1 = 0.9$, $\beta_2 = 0.999$ i.e. we fix $\beta_1, \beta_2$ to be the optimal values from the previous sweep. This performed worse that warmup of 51.2k steps.

3. AdamW constant fraction weight averaging: - learning rate in [3.16e-4, 1e-3, 3.16e-3], $\beta_1 = 0.9$, $\beta_2$ in [0.99, 0.997, 0.999, 0.9997], $\delta$ in [0.05, 0.1, 0.2].

4. AdamW with cosine decay and weight averaging - learning rate in [3.16e-4, 1e-3, 3.16e-3], $\beta_1 = 0.9$, $\beta_2 = 0.999$, $\delta$ in [0.025, 0.05, 0.1].

5. Accelerated SGD based AdamW with cosine decay - learning rate in [3.16e-4, 1e-3, 3.16e-3], $\beta_1$ in [0.999, 0.99968, 0.9999], $\beta_2$ in [0.99, 0.9968, 0.999], $\beta_3 = 0.9$

6. Accelerated SGD based AdamW with constant learning rate and weight averaging - learning rate in [3.16e-4, 1e-3, 3.16e-3], $\beta_1$ in [0.99684, 0.999], $\beta_2$ in [0.999], $\beta_3 = 0.9$, $\delta$ in [0.05, 0.1]

7. Accelerated SGD based AdamW with cosine decay and weight average - learning rate in [3.16e-4, 1e-3, 3.16e-3], $\beta_1$ in [0.99684, 0.999], $\beta_2 = 0.999$, $\delta$ in [0.05, 0.1], $\beta_3 = 0.9$

8. Schedulefree AdamW with constant learning rate - learning rate in [3.16e-4, 1e-3, 3.16e-3, 1e-2], $\beta_1$ in [0.8, 0.9, 0.95], $\beta_2 = 0.999$

9. Schedulefree AdamW with cosine decay - [3.16e-4, 1e-3, 3.16e-3, 1e-2], $\beta_1$ in [0.8, 0.9, 0.95], $\beta_2 = 0.999$

10. MARS - [3.16e-4, 1e-3, 3.16e-3, 1e-2], $\beta_1$ in [0.9, 0.95 0.99], $\beta_2$ in [0.99, 0.999], $\gamma$ in [0.0, 0.01, 0.02, 0.03, 0.04, 0.05], precondition 1d was set to True.

11. AdEMAMix - [3.16e-4, 1e-3, 3.16e-3], $\beta_1$ in [0.0, 0.9], $\beta_2 = 0.999$, $\beta_3$ in [0.99, 0.999, 0.9999], $\alpha$ in [2,4,8,16].

12. Sim-AdEMAMix - [1e-6, 3.16e-6, 1e-5, 3.16e-5], $\beta_1$ in [0.99, 0.999, 0.9999], $\beta_2 = 0.999$, $\alpha$ in [10, 20, 50, 100]

The hyperparameter sweeps for the large batch experiments are provided below:

1. Schedule-Free AdamW: [1e-3, 3.16e-3, 1e-2], $\beta$ in [0.8,0.9,0.95], $\beta_2$ in [0.9,0.95], $r$ in [0.0, 5.0, 9.0, 50.0]

2. AdamW: [1e-3, 3.16e-3, 1e-2], $\beta_1$ in [0.9,0.95], $\beta_2$ in [0.9, 0.95]

3. LAProp: [1e-3, 3.16e-3, 1e-2], $\beta_1$ in [0.9,0.95], $\beta_2$ in [0.9, 0.95]

4. AdEMAMix: [1e-3, 3.16e-3, 1e-2], $\beta_1$ in [0.0, 0.9], $\beta_2 = 0.95$, $\beta_3$ in [0.9, 0.95, 0.99], $\alpha$ in [2,4,8,16]

5. Sim-AdEMAMix: [1e-4, 3.16e-4, 1e-3], $\beta_1$ in [0.9, 0.95, 0.99], $\beta_2 = 0.95$, $\alpha$ in [0.0, 0.5, 1.0]

The overall GPU hours (on a single H100) used for the above sweeps were approximately $10k$.

Regarding the optimal hyper parameters, those are stated below:

1. For the small batch experiments (Figure 1):
   a. AdamW-cosine - $\eta : 1e^{-3}, \beta_1 : 0.9, \beta_2 : 0.999$
   b. AdamW-wavg - $\eta : 1e^{-3}, \beta_1 : 0.9, \beta_2 : 0.999, \delta : 0.05$
   c. AccAdamW-wavg - $\eta : 1e^{-3}, \beta_1 : 0.999, \beta_2 : 0.999, \beta_3 : 0.9, \delta : 0.1$
   d. ScheduleFree-AdamW - $\eta : 3.16e^{-3}, \beta : 0.9, \beta_2 : 0.999$

e. MARS - $\eta : 3.16e^{-3}, \beta_1 : 0.95, \beta_2 : 0.9999, \gamma : 0.1$

f. AdEMAMix: $\eta : 3.16e^{-4}, \beta_1 : 0.9, \beta_2 : 0.999, \beta_3 : 0.9999, \alpha : 16.0$

g. Sim-AdEMAMix: $\eta : 1e^{-6}, \beta_1 : 0.9999, \beta_2 : 0.999, \beta_{\text{start}} : 0.9, \alpha : 100$

2. For the large batch experiments of ScheduleFree (Figure 2):

a. AdamW-cosine: $\eta : 3.16e^{-3}, \beta_1 : 0.95, \beta_2 : 0.95$

b. ScheduleFree-AdamW: $\eta : 1e^{-2}, \beta_1 : 0.9, \beta_2 : 0.9, r : 9.0$

c. LAProp: $\eta : 1e^{-2}, \beta_1 : 0.95, \beta_2 : 0.95$

d. AdamW-wavg: $\eta : 1e - 2, \beta_1 : 0.9, \beta_2 : 0.95, \delta : 0.05$

3. For the large batch experiments of AdEMAMix (Figure 3):

a. AdEMAMix (beta1 = 0.0): $\eta : 1e^{-3}, \beta_1 : 0.0, \beta_2 : 0.95, \beta_3 : 0.95, \alpha : 16.0, \alpha_{\text{start}} : 8.0, \beta_{\text{start}} : 0.9$

b. AdEMAMix: $\eta : 1e^{-3}, \beta_1 : 0.9, \beta_2 : 0.95, \beta_3 : 0.99, \alpha : 8.0$

c. Sim-AdEMAMix: $\eta : 1e^{-4}, \beta_1 : 0.99, \beta_2 : 0.95, \alpha : 0.0, \beta_{\text{start}} : 0.9$

4. For the small batch experiments of 300m:

a. AdamW-cosine: $\eta : 1e^{-3}, \beta_1 : 0.95, \beta_2 : 0.999$

b. AdEMAMix: $\eta : 3.16e^{-4}, \beta_1 : 0.9, \beta_2 : 0.999, \beta_3 : 0.99968, \alpha : 8.0, \alpha_{\text{start}} : 1.0, \beta_{\text{start}} : 0.9$

c. Sim-AdEMAMix: $\eta : 3.16e^{-6}, \beta_1 : 0.999, \beta_2 : 0.999, \alpha : 20.0, \beta_{\text{start}} : 0.9$

5. For the large batch experiments of 300m:

a. AdEMAMix ($\beta_1 = 0.0$): $\eta : 1e^{-3}, \beta_1 : 0.0, \beta_2 : 0.95, \beta_3 : 0.99, \alpha : 16.0, \alpha_{\text{start}} : 8.0, \beta_{\text{start}} : 0.9$

b. AdEMAMix: $\eta : 1e^{-3}, \beta_1 : 0.9, \beta_2 : 0.95, \beta_3 : 0.99, \alpha : 8.0, \alpha_{\text{start}} : 0.0, \beta_{\text{start}} : 0.9$

c. Sim-AdEMAMix: $\eta : 1e^{-4}, \beta_1 : 0.99, \beta_2 : 0.95, \alpha : 0.05, \beta_{\text{start}} : 0.9$

Note that the Hyperparameter $r$ mentioned for schedule-free AdamW can be found on the open source repo and is also mentioned in the paper on line 432.

Also, by default, the AdEMAMix official repository warms up $\alpha$ from 0.0 and $\beta_3$ from the value of $\beta_1$. To generalize the official repository and to remove the confounder of warming up $\beta_3$ from $\beta_1$ (for the $\beta_1 = 0.0$ experiments), we added the hyperparameters $\alpha_{\text{start}}$ and $\beta_{\text{start}}$ for AdEMAMix. The code for AdEMAMix and Simplified AdEMAMix can be found at https://anonymous.4open.science/r/Sim-AdEMAMix-072E.

**Discussion on hyperparameter sweeps:** While we present above the final hyperparameter sweeps, these were carefully selected to ensure that the optimal value for various hyperparameters does not lie at the edge of the sweep. One particular thing to note is that the optimal learning rate for simplified AdEMAMix is smaller than AdEMAMix (this can be seen in the sweep). This is because of the momentum being maintained in theory style in Simplified AdEMAMix (which implicitly scales the learning rate by $\frac{1}{1-\beta}$).

**Discussion on optimal hyperparameters for Simplified AdEMAMix:** The optimizer by default has the momentum maintained in theory style (not EMA style) with bias correction turned off, which generally seems to help in practice with cosine decay. Optimal value of $\alpha$ really depends on the batch size, and from theory, should scale down linearly with increase in batch size. Our optimal alpha at a batch size of 1m tokens was close to 0, while at 32k was close to 100. At higher batch sizes (i.e curvature dominated regime, instead of noise dominated), $\alpha$ should be set close to a small multiple of $1 - \beta_1$ (inspired by Nesterov).

For tuning $\eta, \beta_1, \beta_2$ and $min_{beta}$, if we have an optimal Adam run with hyperparameters $\eta^{adam}, \beta_1^{adam}$ and $\beta_2^{adam}$, we recommend that for SimAdEMAMix, the optimal hyperparameters should be around $min_{beta} = \beta_1^{adam}$, $\beta_1$ higher than $min_{beta}$ (for $min_{beta} = 0.9$, maybe try 0.95, 0.99, 0.999), $\beta_2 = \beta_2^{adam}$ and $\eta = \eta^{adam}\sqrt{(1 - \min \text{ beta}) * (1 - \beta_1)}$ (thus optimal $\eta$ is coupled with value of $\beta_1$). One more thing to note is that $\beta_1$ should generally decrease with increasing batch size.

## B EQUIVALENCE OF PREVIOUS ACCELERATION METHODS

The general accelerated SGD form is provided in Equation (1). In this section, we will show that all the methods in the works Jain et al. (2018); Vaswani et al. (2019); Liu & Belkin (2020); Gupta et al. (2023) fall within this form.

### B.1 AGNES

The update for Gupta et al. (2023) is given below:

$$x'_n = x_n + \alpha v_n \qquad x_{n+1} = x'_n - \eta g'_n \qquad v_{n+1} = \rho_n(v_n - g'_n)$$

where $g'_n$ is stochastic gradient evaluated on $x'_n$ and the final function is evaluated on $x_n$. The above equations can be rewritten as

$$x'_{n+1} = x'_n - \eta g'_n + \alpha v_{n+1} \qquad -\frac{v_{n+1}}{\rho_n} = \rho_{n-1}\left(-\frac{v_n}{\rho_{n-1}}\right) + g'_n$$

Thus $x'_{n+1}$ follows update equation of the form of Equation (1).

### B.2 ASGD

The update for Jain et al. (2018) is given by:

$$y_{j-1} = \alpha x_{j-1} + (1-\alpha)v_{j-1} \qquad x_j = y_{j-1} - \delta g_{j-1} \qquad z_{j-1} = \beta y_{j-1} + (1-\beta)v_{j-1} \qquad v_j = z_{j-1} - \gamma g_{j-1}$$

where $g_{j-1}$ represents the stochastic gradient evaluated on $y_{j-1}$ and the function is evaluated on the tail averaged $x$.

The update equations above can be rewritten as:

$$y_j = y_{j-1} - \alpha\delta g_{j-1} - (1-\alpha)[y_{j-1} - v_j] \qquad \frac{y_{j-1} - v_j}{\gamma - (1-\beta)\alpha\delta} = (1-\beta)\alpha\frac{y_{j-2} - v_{j-1}}{\gamma - (1-\beta)\alpha\delta} + g_{j-1}$$

The update equations above follow the form of Equation (1).

### B.3 MASS

The update for Liu & Belkin (2020) is given by:

$$w_{t+1} = u_t - \eta_1 g_t \qquad u_{t+1} = (1+\gamma)w_{t+1} - \gamma w_t + \eta_2 g_t$$

where $g_t$ is the stochastic gradient evaluated on $u_t$ and the function is evaluated on $w_t$. These equations can be rewritten as:

$$u_{t+1} = u_t - \gamma(w_t - u_t) - [\eta_1(1+\gamma) - \eta_2]g_t \qquad \frac{w_t - u_t}{\eta_1\gamma - \eta_2} = \gamma\frac{w_{t-1} - u_{t-1}}{\eta_1\gamma - \eta_2} + g_t$$

The update equations above follow the form of Equation (1).

### B.4 SGD WITH NESTEROV ACCELERATION

The update for Vaswani et al. (2019) is given by:

$$w_{k+1} = \zeta_k - \eta g_k \qquad \zeta_k = \alpha_k v_k + (1-\alpha_k)w_k \qquad v_{k+1} = \beta_k v_k + (1-\beta_k)\zeta_k - \gamma_k \eta g_k$$

These equations can be rewritten as:

$$\zeta_{k+1} = \zeta_k - \eta g_k + \alpha_{k+1}[v_{k+1} - w_{k+1}]; \qquad v_{k+1} - w_{k+1} = \beta_k(1-\alpha_k)[v_k - w_k] - \eta(\gamma_k - 1)g_k$$

