# OpenReview forum: "Connections between Schedule-Free Optimizers, AdEMAMix, and Accelerated SGD Variants"
_ICLR.cc/2026/Conference — Submitted to ICLR 2026_

### Official Review · Reviewer_DFkx · 2025-10-14

**Soundness:** 2
**Presentation:** 2
**Contribution:** 2
**Rating:** 2
**Confidence:** 4

**Summary:**

The paper connects recent schedule-free optimizers with accelerated SGD, showing that methods like Lion, MARS, and AdEMAMix can be viewed as special cases of accelerated SGD with preconditioning or averaging. Based on this insight, the authors propose a simplified AdEMAMix that removes redundant momentum terms and achieves similar performance on Transformer language models.

**Strengths:**

The paper analyzes the relationship between schedule-free optimizers and accelerated SGD, showing their mathematical equivalence under certain formulations. It derives a simplified version of AdEMAMix that removes one momentum term and compares it empirically on Transformer models, focusing on efficiency and reduced complexity.

**Weaknesses:**

The experimental section is limited, with only a few validation-loss curves presented and no analysis of convergence speed, stability, or downstream task performance.

The experiments focus on medium-sized Transformer models, lacking broader evidence across different architectures or optimization settings.

The claimed equivalence among optimizers is shown mainly through algebraic reformulation without deeper empirical verification.

The proposed simplified AdEMAMix is evaluated only against a narrow set of baselines, making it unclear whether the improvement holds under diverse training regimes.

**Questions:**

Can the authors provide more comprehensive experiments beyond validation-loss curves, such as convergence analysis or downstream performance?

How sensitive is the proposed simplified AdEMAMix to hyperparameters like learning rate or batch size compared to the original version?

Have the authors tested whether the claimed theoretical equivalence holds empirically under different optimizers and training setups?

Would adding comparisons on larger-scale or more diverse tasks change the observed trends?

---

> ### Author Response · Authors · 2025-11-17
>
> We thank the reviewer for their feedback. Before addressing the specific comments of the reviewer, we would like to point out that the code for the optimizer is now available on this repo - https://anonymous.4open.science/r/Sim-AdEMAMix-072E. The optimal hyperparameters have also been mentioned in Appendix A, so the experiments should be easy to reproduce. Our experiments use this publicly available repo - https://github.com/rosieyzh/optimizers-llm.
>
> The comments are addressed below:
>
> 1. **More experiments**: We believe the focus of the paper is on optimization, and thus looking at validation and training loss curves should be sufficient to consider the performance of different optimizers. Moreover, we are in a language modeling setup, where we are using fresh data, thus we consider the validation and training loss curves to look similar. Can the reviewer please convey why looking at the downstream performance is useful in this setup?
>
> 2. **Medium-sized Transformer models**: We would be happy to state that the main focus of the paper is to compare these optimizers on transformer models and language modeling experiments. We have already stated that the current experiments take around 10k H100 GPU hours with the tuning of each optimizer. We believe this amount of computation itself is enough to show the validation at medium-sized transformer models, while leaving the empirical validation at larger scales to future work.
>
> 3. **Equivalence through algebraic reformulation**: We believe this is the strongest form of equivalence between optimizers as it shows they are mathematically precisely the same. Even though the proofs look trivial (and we agree they are), we believe it is a significant contribution to show that many of the existing optimizers are algebraically the same as accelerated SGD. Moreover, for convergence rates, as the optimizers fall directly within the accelerated SGD framework, the convergence rates follow from previous works such as AGNES [1].
>
> 4. **Narrow set of baselines**: We have shown experiments against AdEMAMix, AdamW, MARS and Schedule-Free AdamW. These cover many of the existing diagonal preconditioning optimizers. Note that the focus in the paper is to compare the momentum scheme employed by the optimizers, and thus we are not comparing to non-diagonal preconditioners. Can the reviewer kindly point out which other baselines are important to run?
>
> [1] - Nesterov acceleration despite very noisy gradients, Gupta et al. 2023

---

### Official Review · Reviewer_HLyQ · 2025-10-31

**Soundness:** 2
**Presentation:** 2
**Contribution:** 2
**Rating:** 2
**Confidence:** 3

**Summary:**

This paper establishes a connection between two distinct lines of research: recently proposed optimizers for deep learning and the theory of accelerated stochastic gradient descent. The authors demonstrate that these modern optimizers can be framed as variants of accelerated SGD. Based on this insight authors propose  a new method - Simplified-AdEMAMix, which removes one of the momentum from the AdEMAMix and achieves the same performance in Language Modeling task

Strengthes
1) Insightful Theoretical Connection. The primary strength of this work is the bridge it builds between the practical success of modern optimizers and the formal theory of accelerated SGD. By showing that modern optimizers like MARS and AdEMAMix are mathematically related to specific accelerated SGD formulations, the paper provides a perspective that helps explain why these methods can be effective.
2) Simplified-AdEMAMix. The proposed "Simplified-AdEMAMix" is promissing method. It addresses the significant memory overhead of AdEMAMix, which requires storing three momentum terms. This is particularly relevant for training large models where optimizer state size can be a bottleneck. The empirical results convincingly show that this simplified version maintains the performance of the original AdEMAMix.
3) Strong Empirical Validation. The experiments are conducted on relevant language models (150m and 300m parameters) with substantial training data (up to 15B tokens).

**Strengths:**

1) Insightful Theoretical Connection. The primary strength of this work is the bridge it builds between the practical success of modern optimizers and the formal theory of accelerated SGD. By showing that modern optimizers like MARS and AdEMAMix are mathematically related to specific accelerated SGD formulations, the paper provides a perspective that helps explain why these methods can be effective.
2) Simplified-AdEMAMix. The proposed "Simplified-AdEMAMix" is promissing method. It addresses the significant memory overhead of AdEMAMix, which requires storing three momentum terms. This is particularly relevant for training large models where optimizer state size can be a bottleneck. The empirical results convincingly show that this simplified version maintains the performance of the original AdEMAMix.
3) Strong Empirical Validation. The experiments are conducted on relevant language models (150m and 300m parameters) with substantial training data (up to 15B tokens).

**Weaknesses:**

1) Poorly Supported Claim. The paper hypothesizes that the poor performance of Schedule-Free AdamW at large batch sizes is due to the "coupling between weight averaging and momentum coefficients." While this is a interesting explanation, it is presented without direct evidence. A more convincing approach would involve a targeted ablation study to isolate this coupling effect, rather than just postulating it as the cause for the observed performance gap.
2) The paper claims that "Adam with momentum scheduling can match the performance of AdEMAMix". This is concluded from the observation that the best-performing Simplified-AdEMAMix run corresponds to the case where α=0 in the large batch-size regime, which reduces it to Adam with momentum scheduling. However, this claim raises several questions: why the α=0 is not used in small-batch setting? Why the AdEMAMix does not have this value in the sweep? Does this mean that momentum scheduling can be the real reason, why AdEMAMix performs better than other methods? How Simplified-AdEMAMix performs with α!=0?
3) Hyperparameter Search Spaces. Appendix A reveals that the search spaces for hyperparameters (especially learning rate and α) are vastly different across optimizers. For instance, the learning rate space for Simplified-AdEMAMix ([1e-6, ..., 3.16e-5]) is orders of magnitude different from that for AdEMAMix ([3.16e-4, ..., 3.16e-3]), and their respective α search spaces are non-overlapping. The paper provides no justification for these choices.
4) Simplified-AdEMAMix design. Simplified-AdEMAMix use theory-style momentum, which is uncommon for the DL community. The paper will benefit from discussion, how one should update his AdamW/AdEMAMix hyperparameters to use Simplified-AdEMAMix
4) Lack of Reproducibility: The paper suffers from major reproducibility issues. The authors do not provide source code. Additionally, authors do not report best-performing hyperparameters for each experiment, which with experiment cost (~10k H100 GPU hours as noted in Appendix A) makes reproduction from scratch prohibitively expensive.

**Questions:**

1) The hypothesis regarding the performance degradation of Schedule-Free AdamW due to the coupling of momentum and weight averaging is compelling. Have you considered or run any targeted experiments to isolate and confirm this effect?
2) Could you elaborate on the rationale behind the selection of different hyperparameter search spaces for different optimizers in Appendix A? Specifically, why does Simplified-AdEMAMix require a much smaller learning rate and a different range for α compared to AdEMAMix?
3) Could you please provide the exact best-performing hyperparameters used to generate the plots in Figures 1, 2, and 3? Furthermore, would you consider releasing the source code to allow the community to reproduce and build upon your findings?
4) Why the α=0 is not used in small-batch setting? Why the AdEMAMix does not have this value in the sweep? How Simplified-AdEMAMix performs with α!=0?

---

> ### Author Response · Authors · 2025-11-17
>
> We thank the reviewer for their feedback. We are happy to see that the reviewer finds our theoretical connections insightful. Before addressing the specific comments of the reviewer, we would like to point out that the code for the optimizer is now available on this repo - https://anonymous.4open.science/r/Sim-AdEMAMix-072E. The optimal hyperparameters have also been mentioned in Appendix A, so the experiments should be easy to reproduce. Our experiments use this publicly available repo - https://github.com/rosieyzh/optimizers-llm.
>
> The comments are addressed below:
>
> 1. **Schedule-Free AdamW not working at large batch sizes**: We believe we have removed one confounder concerning the LAProp vs AdamW style preconditioning. Although as the reviewer states, it is still possible that at large batch sizes, maybe constant lr with weight averaging just cannot recover the performance of decay. Thus, we have also added AdamW-wavg at large batch sizes, where we independently sweep momentum and weight averaging coefficient for AdamW and show that it recovers the performance. We think this shows that the coupling of weight averaging and momentum is primarily responsible for the worse performance of Schedule-Free at large batch sizes.
>
> 2. **Adam with momentum scheduling matching AdEMAMix**: The reason for not using $\alpha=0$ in the sweep at small batch size was that the small values were already performing worse at small batch sizes. We have listed now the optimal values for different hyperparameters in the sweeps as well in Appendix A, and as can be seen that the optimal $\alpha$ values lie in between the sweep, so we already saw that smaller alpha values perform worse. This clearly indicates that $\alpha$ is crucial at small batch sizes as expected from the literature on accelerated SGD.
>
> 3. **Hyperparameter search spaces**: The main difference in the learning rate search space is because of the use of theory style momentum vs practical style momentum in Simplified AdEMAMix vs AdEMAMix. Due to this difference, we think the learning rate of simplified AdEMAMix should be scaled down as it is implicitly scaled up by approximately $1/(1-\beta_1)$. This is discussed in the repo - https://anonymous.4open.science/r/Sim-AdEMAMix-072E.
>
> 4. **Reproducibility**: The anonymous code for the method is now available on this repo - https://anonymous.4open.science/r/Sim-AdEMAMix-072E . Moreover the optimal hyperparameters for various runs have now been listed in Appendix A, so the experiments can be easily reproduced.
>
> If our response has addressed the concerns of the reviewer, we would be grateful if the reviewer can update the score accordingly.

---

> > ### Comment · Reviewer_HLyQ · 2025-11-24
> >
> > Thanks for the great effort the authors have been made. I appreciate both new results (experiment with AdamW-wavg) and the commitment to reproducibility (the repo and best hyperparameters). I found the discussion in the repository on hyperparameter selection and practical recommendations for using Simplified AdEMAMix to be very insightful. In my view, this discussion should also be presented in the main paper. Additionally, I recommend authors to include some short discussion on hyperparameters sweeps selection into Appendix. While current sweeps are fair (as authors prove in their response), they can confuse readers  who read the work for the first time. If the authors can add these discussions to the paper, I am ready to reconsider my score (now I give 4).

---

> > > ### Author Response · Authors · 2025-11-26
> > >
> > > Thanks for the response. We have updated the paper and added discussion on hyperparameter sweeps and optimal hyperparameter settings for Simplified AdEMAMix in the Appendix. We would be happy to address any further questions and concerns of the reviewer.

---

> > > > ### Comment · Reviewer_HLyQ · 2025-11-27
> > > >
> > > > Thanks!
> > > >
> > > > For me, this is a borderline paper and __my score for it is 5__. But there is no such rating, so I'll put a 6 for now, so that there are two positive ratings and leave the decision to AC.

---

### Official Review · Reviewer_HVmQ · 2025-10-31

**Soundness:** 2
**Presentation:** 2
**Contribution:** 2
**Rating:** 2
**Confidence:** 3

**Summary:**

The paper proposes a generalized accelerated-SGD framework and establishes connections between several recent optimizers, including Schedule-Free, Lion, MARS, AdEMAMix. The framework decouples the momentum coefficient from current gradient weight. It further proposes Simplified-AdEMAMix that use two momentums instead of three that can achieve comparable performance as AdEMAMix, outperforming other methods in the high batch size regims. Experiments on 150M/300M LLMs are provided to support the claims.

**Strengths:**

1. The unifying view is clear and compact.

2. Experimental details are clearly provided.

**Weaknesses:**

1. Limited theoretical novelty and significance. Much of the connection is pure algebraic rewriting of algorithms, and the unification seems superficial. The paper does not provide formal performance guarantee or new convergence results that explains the different convergence behavior of different methods.

2. The proposed Simplified-AdEMAMix seems incremental. As shown in Algorithm 2, it removes the $(1-\beta_1)$ factor before gradient for first momentum (step 5), and add another gradient term in the update (step 7). It is unclear why this brings significant improvement.

3. Experiments are limited. The size of the model is relatively small. It is unclear whether the claims can be transferred to larger models, e.g., 1B size.

**Questions:**

1. Is there any theoretical justification on why AdEMAMix, "which most closely resembles accelerated stochastic gradient descent", works the best? Why AdamW deviates from this framework?

---

> ### Author Response · Authors · 2025-11-17
>
> We thank the reviewer for their feedback. We are happy to see that the reviewer finds the unifying view clear and compact. Before addressing the specific comments of the reviewer, we would like to point out that the code for the optimizer is now available on this repo - https://anonymous.4open.science/r/Sim-AdEMAMix-072E. The optimal hyperparameters have also been mentioned in Appendix A, so the experiments should be easy to reproduce. Our experiments use this publicly available repo - https://github.com/rosieyzh/optimizers-llm.
>
> The specific comments are addressed below:
>
> 1. **Theoretical Novelty and Significance**: Yes, we completely agree that the proofs shown in the paper depend on pure algebraic rewriting of the algorithms. However, we do believe the results are significant as it shows that the plethora of optimizers being proposed recently are simply instances of accelerated SGD. We think clarifying the space of optimizers is a significant contribution. Moreover, the convergence guarantees for these optimizers directly hold from previous works such as AGNES[1] (considering only the momentum part, not the preconditioning part).
>
>
> 2. **Simplified-AdEMAMix**: The main contribution here is removing the extra memory overhead of AdEMAMix as compared to AdamW. We are basically showing that AdamW can be improved without any memory overhead by using simplified AdEMAMix. We don’t need the extra buffer as required by AdEMAMix. We believe this makes it significantly easier to directly replace AdamW with simplified AdEMAMix.
>
>
> 3. **Limited Experiments** - The experiments already required 10k H100 GPU hours as stated in the appendix. This is a significant cost as we are doing proper sweeps for different optimizers. We would be happy to add an explicit limitations section stating that the experiments are limited to 300m scale.
>
>
> 4. **AdEMAMix closely resembling accelerated SGD**: As stated in section 4.4, the default hyperparameters used in AdEMAMix (with beta_1 = 0.0)  track the optimal theoretical setting as in a recent accelerated SGD paper [1]. AdamW does not fit in this framework as it simply uses the heavy ball or nesterov momentum, and does not fall into the accelerated SGD framework (it does not have an independent weight on the gradient term in the final update).
>
> If our response has addressed the concerns of the reviewer, we would be grateful if the reviewer can update the score accordingly.
>
> [1] - Nesterov acceleration despite very noisy gradients, Gupta et al. 2023

---

### Official Review · Reviewer_psGb · 2025-11-04

**Soundness:** 4
**Presentation:** 4
**Contribution:** 4
**Rating:** 8
**Confidence:** 4

**Summary:**

This paper unifies a handful of recent optimizers (Schedule-Free, AdEMAMix, MARS, Lion) into an accelerated SGD framework. Whereas heavy-ball and Nesterov momentum accelerate deterministic gradient descent; accelerated SGD methods that decouple the momentum term from the gradient term in the optimizer update have been proposed to accelerate in the presence of noise. They show that Schedule-Free SGD can be rewritten as accelerated SGD followed by weight averaging. Lion, Schedule-Free AdamW, AdEMAMix (with beta1=0), and LaProp can be rewritten as preconditioners followed by accelerated SGD.

They perform experiments on 150M and 300M Transformers and note several results that are consistent with their theoretical observations. In particular, they note that compared to AdamW, the accelerated methods perform well at small batch sizes but their advantages diminish at large batch sizes. This aligns with the notion that accelerated SGD methods should improve performance only in the noise-dominated regime. They also note that Schedule-Free AdamW performs especially poorly with large batch sizes and note this could be due to the inherent coupling of the momentum and weight averaging coefficients whose optimal values should deviate at large batch size. They also propose a simplification of AdEMAMix with a single momentum term that improves performance at large batch size.

**Strengths:**

This is an excellent paper. The literature on optimizers tends to proliferate closely related methods leaving a lack of understanding and comparison via empirical results that depend heavily on who had more compute to tune hyperparameters. It is refreshing to read a paper that consolidates a handful of interesting optimizers into one framework that sheds light on empirical results (particularly the diminished performance gains of accelerated SGD methods on large batch sizes compared to AdamW).

Section 4.1 that rewrites Schedule-Free as an accelerated SGD update rule with weight averaging is an especially nice insight.

**Weaknesses:**

- Minor comment: the paragraph in the related work that starts with “Over the years, several optimizers have been proposed … “ would be easier to read if you included the name of each optimizer, e.g. “Chen et al (2023) proposed the Lion optimizer discovered via a genetic search algorithm…”

**Questions:**

N/A

---

> ### Author Response · Authors · 2025-11-17
>
> We thank the reviewer for their feedback. We are happy to see that the reviewer finds the connections shown between different optimizers insightful and agrees with our philosophy that providing a consolidated theoretical foundation for various optimizers proposed recently is useful. We have updated the related work as proposed by the reviewer.

---

### Comment · Area_Chair_xNKe · 2025-11-28

Dear Reviewers,

The discussion phase is now underway, and the authors have finished uploading their responses to reviewers. If you haven't already, please carefully review the authors' responses to understand their perspectives. Engage in thoughtful, constructive discussions with authors, sharing your thoughts and seeking clarifications. Please also update your review or rating if necessary.

It is noted in the guideline that reviewers can leave comments visible to authors **until Dec 2 11:59pm AoE**. Your active participation and contribution to the ongoing discussion are highly encouraged. Thank you very much for your contribution to ICLR.

Best regards,

AC

---

### Meta-Review · Area_Chair_zGQ7 · 2026-01-06

**Summary:**

This paper provides a clear and useful unification of several recent optimizers (Schedule-Free methods, AdEMAMix, MARS, and Lion) under the accelerated SGD framework, which helps explain their empirical behavior in noise-dominated regimes. The derivations are correct and well-presented, and experiments on 150M and 300M language models broadly support the narrative.

However, the main limitation is that the contribution is dominated by elementary algebraic rewritings of existing methods. While informative, these derivations take up most of the paper, whereas the genuinely new contribution, Simplified-AdEMAMix, is only briefly introduced. Several reviewers noted that this proposed method should be the central focus of the paper, with a more thorough derivation, clearer motivation, and stronger theoretical and empirical justification. Concerns about limited theoretical novelty, insufficient ablations for some empirical claims, and restricted experimental scope are only partially addressed in the rebuttal.

Overall, this is a solid unification paper; however, the emphasis on derivations over new algorithmic insights limits its impact in its current form.

**Reviewer Concerns:**

**Addressed:**

* Reproducibility concerns via code and hyperparameter release.
* Clarifications on the relationship between AdEMAMix and accelerated SGD.

**Outstanding:**

* Limited novelty beyond algebraic reformulation.
* Simplified-AdEMAMix is underdeveloped and insufficiently justified.
* Lack of targeted ablations to support key empirical explanations.

**Reviewer Scores:**

It seems to me that there is no clear consensus among reviewers. I believe one reviewer, Reviewer psGb, would support acceptance, while Reviewer HLyQ is borderline, and the last two reviewers are leaning towards rejection.

---

### Decision · Program_Chairs · 2026-01-26

Reject